# Development of a Pharmacogenetic Lab-on-Chip Assay Based on the In-Check Technology to Screen for Genetic Variations Associated to Adverse Drug Reactions to Common Chemotherapeutic Agents

**DOI:** 10.3390/bios10120202

**Published:** 2020-12-09

**Authors:** Rosario Iemmolo, Valentina La Cognata, Giovanna Morello, Maria Guarnaccia, Mariamena Arbitrio, Enrico Alessi, Sebastiano Cavallaro

**Affiliations:** 1Institute for Biomedical Research and Innovation, National Research Council, Via Paolo Gaifami, 18-95126 Catania, Italy; r.iemmolo@isn.cnr.it (R.I.); valentina.lacognata@cnr.it (V.L.C.); g.morello@isn.cnr.it (G.M.); maria.guarnaccia@cnr.it (M.G.); 2Institute for Biomedical Research and Innovation, National Research Council, 88100 Catanzaro, Italy; mariamena.arbitrio@cnr.it; 3Analog, MEMS & Sensor Group Health Care Business Development Unit, STMicroelectronics, Stradale Primosole, 50-95126 Catania, Italy; enrico.alessi@st.com

**Keywords:** pharmacogenetics, lab-on-chip, microfluidics, biosensors, In-Check platform, adverse drug reaction

## Abstract

Background: Antineoplastic agents represent the most common class of drugs causing Adverse Drug Reactions (ADRs). Mutant alleles of genes coding for drug-metabolizing enzymes are the best studied individual risk factors for these ADRs. Although the correlation between genetic polymorphisms and ADRs is well-known, pharmacogenetic tests are limited to centralized laboratories with expensive or dedicated instrumentation used by specialized personnel. Nowadays, DNA chips have overcome the major limitations in terms of sensibility, specificity or small molecular detection, allowing the simultaneous detection of several genetic polymorphisms with time and costs-effective advantages. In this work, we describe the design of a novel silicon-based lab-on-chip assay able to perform low-density and high-resolution multi-assay analysis (amplification and hybridization reactions) on the In-Check platform. Methods: The novel lab-on-chip was used to screen 17 allelic variants of three genes associated with adverse reactions to common chemotherapeutic agents: *DPYD* (Dihydropyrimidine dehydrogenase), *MTHFR* (5,10-Methylenetetrahydrofolate reductase) and *TPMT* (Thiopurine S-methyltransferase). Results: Inter- and intra assay variability were performed to assess the specificity and sensibility of the chip. Linear regression was used to assess the optimal hybridization temperature set at 52 °C (R^2^ ≈ 0.97). Limit of detection was 50 nM. Conclusions: The high performance in terms of sensibility and specificity of this lab-on-chip supports its further translation to clinical diagnostics, where it may effectively promote precision medicine.

## 1. Introduction

Cancer is one of the leading causes of mortality and a major public health problem worldwide [1,2]. Anticancer therapeutic strategies are influenced by tumor clinical characteristics, such as signs, symptoms, histological type, stage and localization. Currently, the most used cytotoxic antineoplastic drugs are pyrimidine analogues (i.e., 5-fluorouracil, capecitabine or tegafur), purine analogues (i.e., mercaptopurine or thioguanine), and platinum compounds (i.e., cisplatin or oxaliplatin). These drugs have a narrow therapeutic index and strictly dose-related effects that are conditioned by inter-individual variability in their metabolism. Therefore, interest in the relationship between genetic variants and cancer treatment outcomes has been growing. Pharmacogenetic (PGx) research in cancer patients has identified specific allelic variants of genes that are related to antineoplastic drug Absorption, Distribution, Metabolism and Excretion (ADME) and can alter pharmacokinetic and pharmacodynamic parameters. This, in turn, results in variable efficacy in tumors of the same histotype [3] or adverse drug reactions (ADRs) [4]. It is now accepted that polymorphic variants in ADME-related genes impact individual patients’ drug sensitivity, resistance and toxicity and contribute to 25–50% of inappropriate pharmacological responses, ranking between the fourth to sixth leading causes of death [5,6]. The most common genetic alterations studied are Single Nucleotide Polymorphisms (SNPs), genomic insertions and deletions, and genetic Copy Number Variations (CNVs). SNPs are a single base-pair difference in the DNA sequence of individuals and represent common inherited variations (90%) distributed throughout the genome. When located within a gene coding sequence or in a regulatory region, they may exert a functional role. SNPs are stably inherited within haplotype blocks in linkage disequilibrium (LD) with a specific gene variant and coheredited within the haplotype. They can be considered as biomarkers of a gene variant and are often used in genomic analyses as tags (tagSNPs) to identify a haplotype block in which few or many polymorphic variants are associated with a disease or drug-response phenotype. The analysis of ADME-related gene polymorphisms can be used for drug efficacy prediction, and screening of individuals who should avoid or receive an adjusted dose of the drug is recommended [7,8,9]. Unfortunately, pharmacogenetic evaluations for individualized antineoplastic drug response are not common in clinical practice for different reasons. The controversial clinical relevance of pharmacogenetic biomarkers, the heterogeneity in efficacy and toxicity of ADRs, the time lag between scientific findings, drug label annotations and clinical practice guidelines are only a few major challenges in the clinical implementation of pharmacogenetics testing [10,11,12]. Moreover, the genetic variations associated with the response to drugs are manifold and therefore their evaluation undoubtedly takes a long time and prohibitive costs. For these reasons, pharmacogenetic tests are not usually implemented in clinical care but often performed only in centralized laboratories with expensive or dedicated instrumentation used by specialized personnel [7,13,14]. In the last few years, several biosensors have been developed with the purpose of promoting precision medicine with advantages in terms of efficiency, time and costs of analysis. However, only few studies report the development of miniaturized DNA chip performing allelic variant discrimination to avoid ADRs in oncological patients [15,16,17,18].

The In-Check platform developed by STMicroelectronics is an innovative technology in the field of microfluidic-based DNA biosensors that combines micro-electro-mechanical-system (MEMS) techniques. The platform consists of three main modules: the Temperature Control System (TCS) that operates and monitors the thermal cycling reactions, allowing a fast temperature ramping with an accuracy of 0.1 °C with improved performance and amplification time; the portable Optical Reader (OR) to perform scanning of the chip; a miniaturized Lab-on-Chip (LoC) device that brings together silicon-based microelectronics with micromachining technology. LoC consists of two separate Polymerase Chain Reaction (PCR) chambers and a microarray area joined together through microfluidic channels allowing a fast, highly sensitive and specific amplification of nucleic acids and competitive hybridization of generated amplicons simultaneously (Figure 1). 

The miniaturized low-density microarray area of the In-Check LoC contains up to 126 spots, offering the opportunity to design and customize the chip for individual applications. The platform is completed with dedicated software that controls the instruments and runs the data analysis [19,20,21]. The miniaturization of the combined PCR-microarray analytical processes performed into a unique biosensor allows different advantages in terms of high sensitivity, simplicity and high-throughput, with short assay time, low reagent consumption, and easiness to automation [22]. Today, the platform is marketed by Sekisui—Veredus Ltd. and is used with a series of customized panels for infection disease, food safety and biosurveillance (vereduslabs.com). In this work, we describe the development of PGx Lab-on-Chip test based on In-Check technology, which is able to detect multiple allelic variants of three genes (*DPYD*, Dihydropyrimidine dehydrogenase; *MTHFR*, 5,10-Methylenetetrahydrofolate reductase; and *TPMT*, Thiopurine S-methyltransferase) associated to adverse reactions to common chemotherapeutic agents.

## 2. Materials and Methods

### 2.1. Development of a Pharmacogenetic Lab-On-Chip

Our custom PGx Lab-on-Chips were developed for the In-Check technology and fabricated as previously described [20]. We designed 31 allele-specific capture probes able to perfect match the selected wild type and 9 different mutant allelic variants of *DPYD*, *MTHFR* and *TPMT* genes (Table 1). An additional artificial oligonucleotide (AT683) was designed and used both as spike-in internal hybridization control and for grid alignment. Purified allele-specific capture probes and hybridization controls were modified with an amino group and a six carbon spacer at their 5′-terminus and spotted onto silicon microarray slide as previously described [19].

### 2.2. Asymmetric Multiplex PCR and Gene Sequencing

In-Check LoCs can simultaneously perform two asymmetric multiplex PCR reactions in two different PCR chambers. The multiplex PCR design, primer pair sequences and sizes of amplicons are listed in Table 2. PCR mixes were set up as follows: 2.5 U of HotStart Taq (Qiagen, Hilden, Germany), 0.6 mM MgCl_2_, 0.56 mM of each dNTP, 2.5 µl of 10× PCR buffer, 0.2 µM of forward primer, 2.4 µM of Cy5-labeled reverse primer and 50 ng of standard genomic DNA in a final volume of 12.5 µL. Then, 11.5 µl of each PCR mix was dispended into the two chamber inlets of the silicon lab-on-chips which were closed with specific clamps and loaded into Temperature Control System (TCS) sets as follow: 95 °C × 300″–35 cycles at 94 °C × 60″, 61 °C × 60″, 72 °C × 60″–72 °C × 600″. To assess the accuracy of multiplex PCRs, amplicons were recovered from the PGx-LoC by brief centrifugation in 50 mL Falcon tubes, and then analyzed on 2% TBE agarose gel and visualized under UV transilluminator. Amplicons were also directly sequenced with unconjugated primer pairs using BigDye terminator v.3.1 kit (Applied Biosystems, Foster City, CA, USA) according to manufacturer’s instructions, on ABIPRISM 310 genetic analyzer (Applied Biosystems, Foster City, CA, USA). GeneBank sequences for *DPYD* (#NG_008807.2 RefSeqGene), *MTHFR* (#NG_013351.1 RefSeqGene) and *TPMT* (#NG_012137.2 RefSeqGene) genes were used for multiple alignments using CLC Sequence Viewer 7.6.1 software (CLC bio, Cambridge, MA, USA).

### 2.3. Microarray Hybridization and Scanning

The next step in the In-Check assay workflow is the microarray hybridization of PCR products. To allow migration of PCR products to the Microarray area, 14.5 μL of hybridization mix (a mixture of 2x hybridization buffer and 500 nM spike-in Cy5 labeled hybridization control) was pumped into PCR chambers. Then, PGx-LoCs were loaded into TCS for 30 min. The hybridization temperature plays a crucial role in hybridization-based genotyping assay. In order to calculate the accuracy of allele-specific capture probes, different hybridization temperatures were tested (50 °C, 52 °C, 54 °C, 56 °C). Three replicates for each hybridization test were performed. After hybridization, PGx-LoCs were centrifuged for 2 min at 3000 rpm into a 50-mL falcon tubes in stringency washing buffer (2× SSC + 0.1% SDS) at room temperature and dried through a second centrifugation for 2 min at 3000 rpm into empty 50 mL falcon tubes. Finally, PGx-LoCs were scanned with a dedicated Optical Reader to acquire fluorescence intensities. Raw signal values were background subtracted and normalized by customized software.

### 2.4. Statistical Analysis

The custom LoC assay described here was analyzed for its repeatability, sensibility and specificity by studying the Limit of Blank (LoB), Limit of Detection (LoD), and inter- and intra-assay variability as previously reported [22]. For each fluorescent image, the signal intensity for each spot signal was determined by subtracting background intensity from the mean average intensity while the target mean values were used for scaling normalization. The relationship between signal standard deviation and the signal median was evaluated using Pearson correlation coefficient values indicated by R^2^. *p*-value ≤ 0.05 was considered significant. A specific software, licensed by STMicroelectronics, was used to perform grid alignment and data analysis.

## 3. Results

### 3.1. Design of a Pharmacogenetic LoC Assay for the In-Check Platform

The In-Check platform has immense potentiality in medical diagnostics. It satisfies the cornerstones of biosensors: simplicity in operation, higher sensitivity and the ability to perform multiplex analysis [23]. The functional unit of the In-Check platform is a silicon-based lab-on-chip able to perform low-density multigenic analysis (targets amplification and detection by hybridization) in less than 2 h, with cost-effective and time-consumption advantages. It contains two separate and miniaturized PCR chambers in which two multiplex PCRs can be performed simultaneously (Figure 1). 

The microfluidic connection allows samples to naturally migrate from PCR to Microarray chambers. In this area, up to 126 customized allele-specific capture probes can be spotted. Other components of the platform are the Temperature Control System, the Optical Reader and a complete suite of software modules to easily perform image analysis and data interpretation for the management of diagnostic results [20,24].

In this work, we selected 17 different allelic variants, eight of which are nonfunctional variants, for *DPYD* (g.226525A; g.226525G; g.410273T; g.410273G; g.410273A; g.475992T; g.475992C; g.476002G; g.476002A; g.843669A; g.843669T), *MTHFR* (g.16685A; g.16685C) and *TPMT* (g.16420G; g.16420C; g.21147G; g.21147A) genes (Table 1), based on their involvement in drug response according to PharmGKB (http://www.pharmgkb.org), Online Mendelian Inheritance in Man (OMIM—http://www.omim.org) and Single Nucleotide Polymorphism (dbSNP—http://www.ncbi.nlm.nih.gov/SNP) databases. For the analysis of these variants, primer sets were designed to amplify selected regions with a range of melting temperature of 59.4 ± 1.8 °C, guanine-cytosine (GC) content between 36–55%, and target size from 177 bp to 300 bp (Table 2). In addition, 31 allele-specific capture probes were designed to complementary align with wild type (WT) and mutated (M) sequences (Table 3) subdivided as follows: 22 capture probes complementary to *DPYD* hotspots (10 WT and 12 M); 4 capture probes complementary to *MTHFR* hotspots (2 WT and 2 M), 5 capture probes complementary to *TPMT* hotspots (2 WT and 3 M). After in silico testing, allelic capture probes were synthetized and spotted with a duplicated layout on the PGx-LoC microarray area. Additional capture probes complementary to AT683 hybridization control were spotted with a specular pattern for grid alignment (Figure 2).

### 3.2. Multiplex PCR Optimization

PGx-LoC was designed for a rapid and multiassay system where few manual operations are required. As previously reported, the first step in the genotyping assay is the optimization of PCR performances in order to amplify and detect specific target site variants in genomic DNA with high fidelity [22]. To implement the protocol on the In-Check platform, primers were tested in combination and two multiplex PCR were performed with an optimal annealing temperature of 61 °C. The specificity of conventional real-time PCR versus PGx-LoC was evaluated through direct sequencing in order to confirm the presence or absence of the target amplicons in the mix, due to the inability to exactly discern them on agarose gel electrophoresis (Figure 3).

### 3.3. Analitical Variability of PGx-LoC

In an microarray-based detection assay, different hybridization conditions, such as temperature, denaturation mode, incubation time or the concentration of the analyte, are crucial [25,26]. As reported in our previous study [22], four different hybridization temperatures based upon the temperature melting (Tm) values of allele-specific capture probes, from 50 °C to 56 °C, were tested to evaluate inter- and intra-assay variability. After genomic amplification and hybridization of the selected 17 allelic variants of the wild-type genome shown in Figure 3, fluorescence intensities of allele-specific capture probes were detected and linear regression analysis between median fluorescence intensities and their standard deviations was made. As shown in Figure 4, hybridized wild-type probes were 62% at 50 °C with a R^2^ = 0.96, 100% at 52 °C with a R^2^ ≈ 0.97, 56% at 54 °C with a R^2^ = 0.9147, and 50% at 56 °C with a R^2^ = 0.9405. In particular, at 50 °C, 54 °C and 56 °C no fluorescence was detected for the *DPYD* g.226525A, *DPYD* g.410273T, *DPYD* g.843669A and *TPMT* g.21147G related capture probes and no signal was detected for the mutated capture probes. Both fluorescence intensity and linear regression analysis demonstrated that the signal values for wild-type targets were complete and more accurate when hybridization is performed at 52 °C. At this temperature, the hybridization of 100% wild type probes and no hybridization signal in the mutated probes was observed, indicating this as the optimal hybridization temperature (*p* < 0.0001; R^2^ = 0.9659; Figure 4).

Once the optimal hybridization temperature was identified, the PGx-LoC sensibility was measured by testing LoB and LoD. For this purpose, we randomly chose 4 capture probes complementary to two different SNPs of *DPYD* gene (rs17376848_T and rs3918290_G). As reported in other studies with lab-on-chip assays performed by the In-Check platform [22], the measure of LoB in PGx-LoC is a parameter particularly crucial in a test to demonstrate its sensibility in the detection of non-analytical signal when sample without analyte is analyzed. As reported in Figure 5A, the LoB test of PGx-LoC demonstrated its sensibility in blank assay. To evaluate the LoD, *DPYD* exon 15 was preamplified in PGx-LoC to a final concentration of 500 nM in PCR chamber. After the recovery of PCR product, 10-fold serial dilutions were made to 50 nM and 5 nM and hybridized in the array area of the chip. At the concentration of 5 nM, no fluorescence signal was detected, resulting in a microarray image similar to blank (Figure 5B). At the concentration of 50 nM, a fluorescence signal was detected for all the four allelic-capture probes complementary to the selected hotspots (green rings in Figure 5C), revealing it as the lowest quantity of analyte detectable by the capture probes (Figure 5D).

Once the PCR and hybridization conditions were set, complete experiments of pharmacogenetic genotyping through the use of PGx-LoC could be performed (Figure 5E). 

## 4. Discussion

Despite the development of biological agents, chemotherapy still represents a first line pharmacological strategy for different kinds of solid tumors treatments. It is not surprising that patients with the same histological tumor and/or stage disease have different pharmacological responses, sometime toxic. This occurs because in patients two genomes co-exist: the germline genome of the host and the somatic genome of tumor [3,27]. Different germline and somatic polymorphisms in ADME-related genes may significantly affect drug disposition and tolerance, leading to toxicity [28]. To avoid these unpleasant complications, the pharmacogenetic analysis may be conducted on patient’s genomes. Unfortunately, pharmacogenetic evaluations for individualized antineoplastic drug response are limited in the clinical practice due to the myriad of PGx variants without unclear functional roles [13].

In this article, we describe the development of an innovative silicon-based lab-on-chip able to perform low-density and high-resolution multi-assay analysis (amplification, hybridization and detection) of three genes (*DPYD*, *MTHFR,* and *TPMT*) associated with adverse reactions to common chemotherapeutic agents.

*DPYD* converts dihydrouracil and dihydrothymine in uracil and thymine, respectively, through the reduction of NADP. *DPYD* deficiency leads to an autosomal recessive disease associated with a variable clinical phenotype. Pharmacogenetic research defined *DPYD* as a predictive marker of Fluoropyrimidines (FL) response because it is the first and rate-limiting enzyme of FL catabolic pathway [29]. Based on the clinical relevance of *DPYD* polymorphisms, the Clinical Pharmacogenetics Implementation Consortium (CPIC) recommends at least a 50% reduction in starting dose followed by ulterior reduction of dose based on toxicity for heterozygous patients, and to select an alternate antineoplastic drug for homozygous ones [30]. We have selected five different polymorphisms in *DPYD* involved not only in FL (fluorouracil, capecitabine and tegafur) metabolism but also in the response to four additional antineoplastic drugs: oxaliplatin and leucovorin, cetuximab and bevacizumab. Three polymorphisms cause a change in the amino acid sequence (rs2297595, rs55886062, rs67376798), one is a synonymous mutation (rs17376848), and one leads to exon 14 skipping (rs3918290). The resulting products are not functional at a different level and may cause severe toxicity. Fortunately, toxic death after FL administration is extremely uncommon, even if it has been hypothesized that the simultaneous presence of more than one allelic *DPYD* variant could exacerbate the toxicity [31]. Even if associated to likely-benign phenotypes, two mutations were recently recommended to be analyzed: rs56038477 (c.1236G > A) and rs1801160 (c.2194G > A) on exons 12 and 14 respectively [32,33,34]. 

The enzyme encoded by *MTHFR* converts irreversibly 5,10-methylenetetrahydrofolate to 5-methyltetrahydrofolate. The latter donates the methyl group to homocysteine to form methionine. In turn, methionine is converted to S-adenosylmethionine (SAM), the methyl donor in DNA methylation [35]. *MTHFR* polymorphisms predispose to serious bone marrow toxicity during treatment with folate synthesis inhibitor [36,37]. We selected the rs1801131 polymorphism that causes the Glu-Ala mutation associated with the reduction of enzyme activity. In tumor tissue, 5-methyltetrahydrofolate is required for optimal fluoropyrimidine efficacy, so that tumors with mutated *MTHFR* should be more sensitive to 5-fluorouracil (5FU) than wild-type tumors. However, *MTHFR*-rs1801131 polymorphism is significantly associated with a worst prognosis in homozygous mutated patients *versus* homozygous wild type patients (2.48 fold relative risk of death) rendering it a significant predictor for 5FU response in colorectal cancer patients [38,39].

TPMT catalyzes the S-methylation of aromatic and heterocyclic sulfhydryl compounds, including the thiopurine drugs 6-mercaptopurine (6MP) and 6-thioguanine (6TG). We selected two SNPs (rs1800462 and rs1800460), present in TPMT 3*A, 3*B and 3*C haplotype. These two SNPs result in lower TPMT protein quantity and also affect TPMT enzymatic activity. Patients carrying these point mutations can develop thiopurine drugs-related toxicity, such as myelosuppression [40]. Based on their genotypes, patients can be separated in slow, intermediate and extensive metabolizers and may suffer from several side effects (reduced enzymatic activities) or have lower therapeutic efficacy (rapid enzymatic activities). CPIC recommends different doses of thioguanine and mercaptopurine ins TPMT haplotypes. In particular, CPIC suggests to start with drastically reduced doses (reducing daily dose by 10-fold and reducing frequency to thrice weekly instead of daily) for poor metabolizers (homozygous mutant variants with two nonfunctional alleles) [41]. 

The intra-assay variability, sensibility and specificity of the silicon-based lab-on-chip analysis described in this study supports its further translation to clinical diagnostics, where it may promote precision medicine by reducing adverse drug reactions to common chemotherapeutic agents.

## 5. Conclusions

One of the challenges of modern medicine is to understand the “costs” of a particular therapeutic approach, in terms of the patient’s quality of life and benefits against the disease. Clinical treatment of oncological patients is made difficult by the adverse reactions of antineoplastic drugs, often linked to genetic factors. The acquired knowledge of the human genome has made possible the development of clinical pharmacogenetic tests in oncological patients. In particular, the design of reliable and low-cost miniaturized diagnostic devices, overcoming the limitations of the current technologies, may facilitate the pharmacogenetic screening of oncological patients, improve prevention of ADRs and promote precision medicine. In this work, we report the development of a novel In-Check LoC able to detect multiple allelic variants in *DPYD* (Dihydropyrimidine dehydrogenase), *MTHFR* (5,10-Methylenetetrahydrofolate reductase) and *TPMT* (Thiopurine S-methyltransferase) which are associated with adverse reactions to common chemotherapeutic agents. The inter- and intra-assay variability testing performed to set the sensitivity and specificity of PCR-microarray combined module, and efficiency of multiplex PCRs and hybridization probes, revealed and an optimal hybridization temperature of 52 °C with the 100% of alleli-capture probes signals were a wild-type genome was analyzed (*p* < 0.0001; slope = 0,2389 to 0,2559; R^2^ = 0.9659). The high performance of the molecular assay, in terms of sensibility, specificity and efficiency, may enable the platform for diagnostic applications. In addition to technical advances, the miniaturized LoC promises cost-effectiveness, short analysis time, low sample/reagent consumption, and ease of use. In summary, the results reported in this study support the translation of PGx-LoC to clinical diagnostics as point-of-care or first-line pharmacogenetics genotyping array for cancer patients. 

## Figures and Tables

**Figure 1 biosensors-10-00202-f001:**
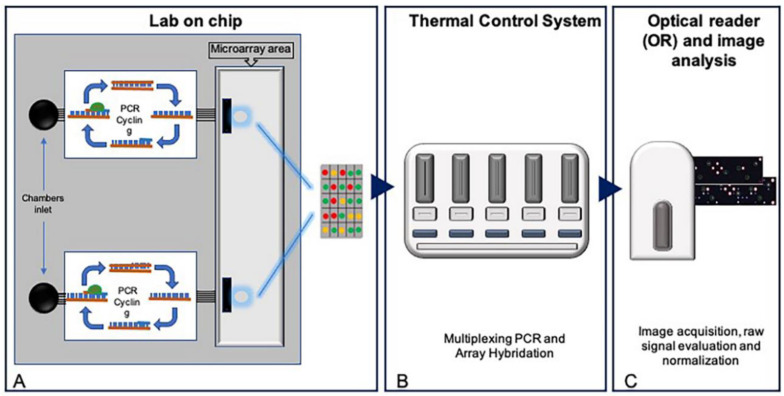
In-Check platform. (**A**) Schematic representation of miniaturized Lab-on-Chip (LoC) device. Two separate PCR chambers and a microarray area are joined together through a microfluidic channel. (**B**) Temperature Control System (TCS) to operate and monitors the reactions thermal cycling. (**C**) Portable Optical Reader (OR) to perform scanning of the chip.

**Figure 2 biosensors-10-00202-f002:**
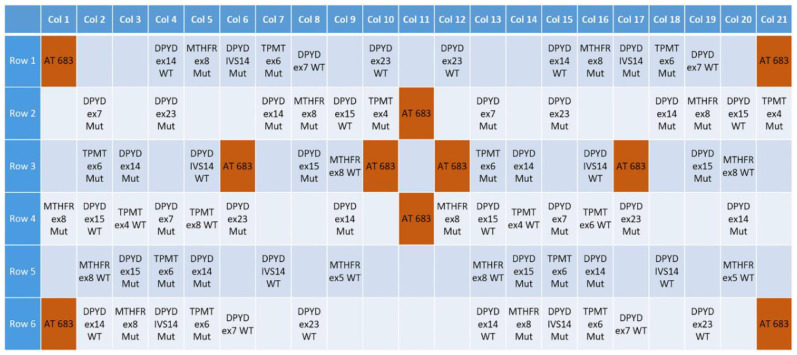
Grid layout of allelic capture and hybridization control probes spotted on the chip array. The overlapping of the virtual grid allows to select each probe cell and test the validity of the call.

**Figure 3 biosensors-10-00202-f003:**
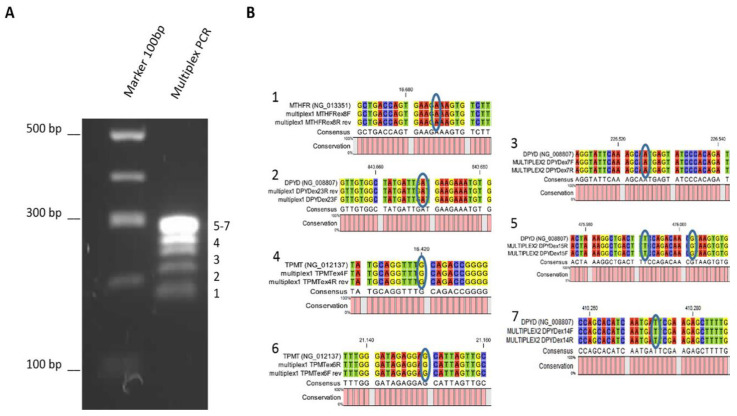
Multiplex PCR performed by the In-Check platform. (**A**) A 2% agarose gel electrophoresis of the multiplex PCR (marker 100bp). On the right, the numerical series identifies the seven amplicons sorted by their lengths. (**B**) Direct sequencing alignments of amplicons. Blue rings indicate selected polymorphisms in MTHFR, DPYD and TPMT genes.

**Figure 4 biosensors-10-00202-f004:**
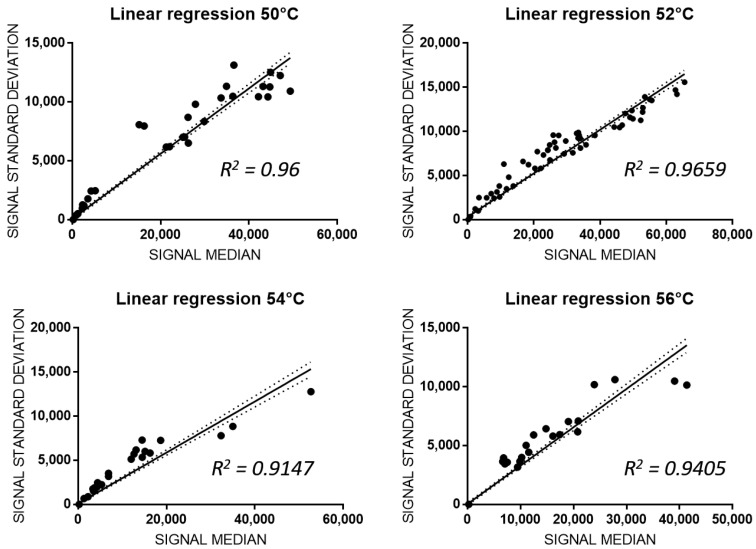
Effects of the hybridization temperature on allele discrimination and SNPs detection. Four different hybridization temperatures were tested and the Linear Regressions of normalized standard deviation and median of fluorescent signals are reported for 50 °C (*p* < 0.0001; slope = 0.2669 to 0.288; R^2^ = 0.96), 52 °C (*p* < 0.0001; slope = 0.2389 to 0.2559; R^2^ = 0.9659), 54 °C (*p* < 0.0001; slope: 0.273 to 0. 3057; R^2^ = 0.9147) and 56 °C (*p* < 0.0001; slope: 0.311 to 0.3414; R^2^ = 0.9405). Best fit in solid lines, 95% intervals of confidence in dashed lines.

**Figure 5 biosensors-10-00202-f005:**
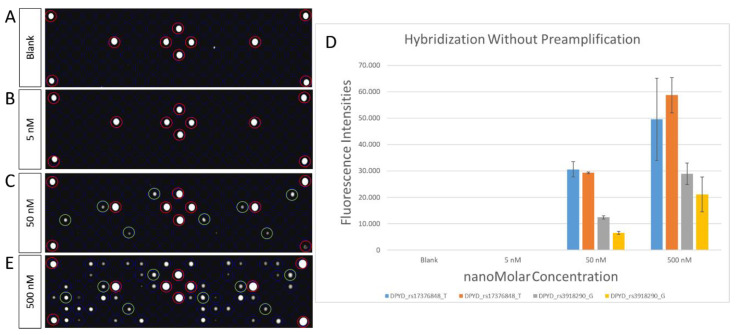
PGx-LoC sensibility measured by Limit of Blank and Limit of Detection. (**A**) Microarray image of Limit of Blank (LoB) with no analyte in the sample. The image reveals only the hybridization control highlighted in red rings. (**B**,**C**) Microarray images of Limit of Detection (LoD) with 10-fold serial dilution of DPYD exon 15 amplicons preamplified in PGx-LoC: 5 nM of PCR product (**B**); 50 nM of PCR product (**C**). Hybridization control highlighted in red rings. Probes of interest highlighted in green rings. (**D**) Fluorescence intensities of blank (0 nM), 5 nM, 50 nM and 500 nM of PCR products hybridized without preamplification relative to the four allelic capture probes chosen for the sensitivity testing. Data are reported as mean ± SD of three independent experiments. (**E**) Microarray image of a complete experiment of genotyping performed with a PGx-LoC with 500 nM of PCR product. Hybridization control highlighted in red rings. Allelic-capture probes complementary to DPYD exon 15 highlighted in green rings.

**Table 1 biosensors-10-00202-t001:** List of allelic variants detected by the pharmacogenetic Lab-on-Chip and associated with Adverse Drug Reactions (ADRs) to antineoplastic drugs.

Gene	Allelic Variants (hg38)	MAF (1000 Genomes)	Antineoplastic Drug	Level(s)
DPYD (NG_008807.2)	rs2297595 g.226525A > G p.Met166Val	C = 0.0565/283	Cetuximab, fluorouracil, capecitabine, oxaliplatin, bevacizumab	2A
	rs55886062 g.410273T > A g.410273T > G p.Ile560Asn; p.Ile560Ser	C = 0.0002/1	Capecitabine, fluorouracil, Pyrimidine analogues, tegafur	1A
	rs17376848 g.475992T > C p.Phe632Phe	G = 0.0521/261	Leucovorin, fluorouracil, capecitabine, oxaliplatin	3
	rs3918290 g.476002G > A IVS14	T = 0.0030/15	Capecitabine, fluorouracil, Pyrimidine analogues, tegafur	1A
	rs67376798 g.843669A > T p.Asp949Val	A = 0.0022/11	Cetuximab, oxaliplatin, bevacizumab, leucovorin, tegafur, fluorouracil, capecitabine, Pyrimidine analogues	1A
MTHFR (NG_013351)	rs1801131 g.16685A > C p.Glu429Ala	G = 0.2494/1249	Leucovorin, capecitabine, fluorouracil, oxaliplatin, methotrexate, bevacizumab, carboplatin, cisplatin, cyanocobalamin, folic acid, or pemetrexed	3
TPMT (NG_012137.1)	rs1800462 g.16420G > C p.Ala80Pro	G = 0.0022/11	s-adenosylmethionine, purine analogues, mercaptopurine, azathioprine, thioguanine	1A
	rs1800460 g.21147G > A p.Ala154Thr	T = 0.0128/64	s-adenosylmethionine, mercaptopurine, purine analogues, azathioprine, thioguanine, cisplatin	1A/3

**Table 2 biosensors-10-00202-t002:** Primer pairs used in multiplex PCR. All reverse primers were labeled with Cy5 at 5′ position.

mPCR	Region of Interest	Primer F	GC %	TM	Primer R	GC %	TM	Product Length
#1	MTHFR exon8	TTTGGGGAGCTGAAGGACTAC	52	61.2	CACTCCAGCATCACTCACTTT	48	59.5	177
	DPYD exon23	TGCAGTACCTTGGAACATTTGG	45	60.1	TGCAGAAGAGCAATATTTGGCA	41	58.4	245
	TPMT exon4	GATCTGCTTTCCTGCATGTTC	48	59.5	TCCAGGAATTTCGGTGATTGG	48	59.5	269
	TPMT exon6	GGACGCTGCTCATCTTCTTA	50	58.4	GACAAAGCTAGTATTGGATTTAGGT	36	60.9	295
#2	DPYD exon7	ACTGAAAATGTACTGCTCATTGCT	38	60.3	CCCCAATCGAGCCAAAAAGG	55	60.5	265
	DPYD exon15	TGTTTCCCCCAGAATCATCCG	52	61.2	TGCATCAGCAAAGCAACTG	47	55	287
	DPYD exon14	AGAAATGGCCGGATTGAAGT	45	56.4	GACAGAAAGGAAGGAAAGAAACTAA	36	60.9	300

**Table 3 biosensors-10-00202-t003:** Allelic specific capture probe sequences designed to genotype ADME-related genes.

Probe Name	Probe Sequences (5′-3′)	5′-End Modification	Probe Type
DPYD_ g.226525A	*TTTTTTT*GGTATTCAAAGCAATGAGTA	5′-C6-NH2	CAPTURE PROBE
DPYD_ g.226525A	*TTTTTTT*AGGTATTCAAAGCAATGAGT	5′-C6-NH2	CAPTURE PROBE
DPYD_ g.226525G	*TTTTTTT*GGTATTCAAAGCAGTGAGTA	5′-C6-NH2	CAPTURE PROBE
DPYD_ g.226525G	*TTTTTTT*AGGTATTCAAAGCAGTGAGT	5′-C6-NH2	CAPTURE PROBE
DPYD_ g.410273T	*TTTTTTT*CATCAATGATTCGAAGAGCT	5′-C6-NH2	CAPTURE PROBE
DPYD_ g.410273T	*TTTTTTT*CACATCAATGATTCGAAGAG	5′-C6-NH2	CAPTURE PROBE
DPYD_ g.410273G	*TTTTTTT*TGAGTCGAAGAGCTTTTGAA	5′-C6-NH2	CAPTURE PROBE
DPYD_ g.410273G	*TTTTTTT*AATGAGTCGAAGAGCTTTTG	5′-C6-NH2	CAPTURE PROBE
DPYD_ g.410273A	*TTTTTTT*TGAATCGAAGAGCTTTTGAA	5′-C6-NH2	CAPTURE PROBE
DPYD_ g.410273A	*TTTTTTT*AATGAATCGAAGAGCTTTTG	5′-C6-NH2	CAPTURE PROBE
DPYD_ g.475992T	*TTTTTTT*TAAAGGCTGACTTTCCAGAC	5′-C6-NH2	CAPTURE PROBE
DPYD_ g.475992T	*TTTTTTT*GAACTAAAGGCTGACTTTCC	5′-C6-NH2	CAPTURE PROBE
DPYD_ g.475992C	*TTTTTTT*TAAAGGCTGACTTCCCAGAC	5′-C6-NH2	CAPTURE PROBE
DPYD_ g.475992C	*TTTTTTT*GAACTAAAGGCTGACTTCCC	5′-C6-NH2	CAPTURE PROBE
DPYD_ g.476002G	*TTTTTTT*TTCCAGACAACGTAAGTGTG	5′-C6-NH2	CAPTURE PROBE
DPYD_ g.476002G	*TTTTTTT*CTTTCCAGACAACGTAAGTG	5′-C6-NH2	CAPTURE PROBE
DPYD_ g.476002A	*TTTTTTT*TTCCAGACAACATAAGTGTG	5′-C6-NH2	CAPTURE PROBE
DPYD_ g.476002A	*TTTTTTT*CTTTCCAGACAACATAAGTG	5′-C6-NH2	CAPTURE PROBE
DPYD_ g.843669A	*TTTTTTT*GGCTATGATTGATGAAGAAAT	5′-C6-NH2	CAPTURE PROBE
DPYD_ g.843669A	*TTTTTTT*GTGGCTATGATTGATGAAGAA	5′-C6-NH2	CAPTURE PROBE
DPYD_ g.843669T	*TTTTTTT*GGCTATGATTGTTGAAGAAAT	5′-C6-NH2	CAPTURE PROBE
DPYD_ g.843669T	*TTTTTTT*GTGGCTATGATTGTTGAAGAA	5′-C6-NH2	CAPTURE PROBE
MTHFR_ g.16685A	*TTTTTTT*CAGTGAAGAAAGTGTCTTTG	5′-C6-NH2	CAPTURE PROBE
MTHFR_ g.16685A	*TTTTTTT*CCAGTGAAGAAAGTGTCTTT	5′-C6-NH2	CAPTURE PROBE
MTHFR_ g.16685C	*TTTTTTT*GTGAAGGAAGTGTCTTTGAA	5′-C6-NH2	CAPTURE PROBE
MTHFR_ g.16685C	*TTTTTTT*CAGTGAAGGAAGTGTCTTTG	5′-C6-NH2	CAPTURE PROBE
TPMT_ g.16420G	*TTTTTTT*GTTTGCAGACCGGGGACA	5′-C6-NH2	CAPTURE PROBE
TPMT_ g.16420C	*TTTTTTT*GTTTCCAGACCGGGGACA	5′-C6-NH2	CAPTURE PROBE
TPMT_ g.21147G	*TTTTTTT*GGATAGAGGAGCATTAGTTG	5′-C6-NH2	CAPTURE PROBE
TPMT_ g.21147A	*TTTTTTT*ATAGAGGAACATTAGTTGCC	5′-C6-NH2	CAPTURE PROBE
TPMT_ g.21147A	*TTTTTTT*GGGATAGAGGAACATTAGTT	5′-C6-NH2	CAPTURE PROBE
AT683	AGTGAGGGAGGAGATGGAACCATCT	5′-C6-NH2	hybridization control

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
