# Peer review of "Development of a Pharmacogenetic Lab-on-Chip Assay Based on the In-Check Technology to Screen for Genetic Variations Associated to Adverse Drug Reactions to Common Chemotherapeutic Agents"

_biosensors, 2020, doi:10.3390/bios10120202_

Round 1

Reviewer 1 Report

This article describe a silicon-based lab-on-chip able to perform multi-assay analysis (amplification, hybridization and detection). Allelic variants of three genes associated to adverse reactions to common chemotherapeutic agents were screened on this platform. There are some problems in the article. The details are as follows:

  1. Since this article describe the development of an innovative silicon-based lab-on-chip able to perform low-density and high-resolution multi-assay analysis based on In-Check technology, the advantages and current status of this technology need to be introduced in the introduction.
  2. The platform this article used include a silicon-based lab-on-chip, temperature control system and the optical reader, and the silicon-based lab-on-chip contains PCR chambers, microarray chambers and microfluidic connection. Such a complex structure of the platform may require a schematic diagram to make it easier for the reader to understand.
  3. The results are not well described, for example, in Figure 4, Panel B, which areas are the results of 0, 5, 50 and 500 nM of PCR product should be marked in the Figure, and the corresponding results should be described in detail in the results section
  4. The conclusion section is too simple to summarize the advantages and characteristics of the diagnostic device used in this article.

Author Response

We have revised the manuscript as follow:

  1. As suggested, the advantages and current status of this technology are added in the introduction.
  2. A schematic discussion of the In-Check modules is now reported in the introduction.
  3. The results were revised and the new figure 1 has been made based on the suggestions of the Reviewer.
  4. As suggested, the conclusion section was re-edited.

Reviewer 2 Report

Dear authors,

Please update the manuscript with the comments suggested below to improve the quality of the manuscript.

  1. In the abstract please mention the limitations of the current/previous studies performed by the peers in this field. Though in lines 20 -22, the authors have stated that the pharmacogenetic tests are limited due to prohibitive costs and lack of necessary equipment, but the limitations with respect to the current technologies are not clearly stated. Please include the details of the technical limitations and specify the advances made to overcome those limitations.
  2. The abstract needs to be incorporated with the gist of the complete work in the manuscript. In the current version of the abstract, only the summary of the work is mentioned in a very broad view. Instead please include specific details of the research work presented in the manuscript.
  3. In the abstract, kindly incorporate the quantification of results to have the reader understand the magnitude of the results and to support the conclusion driven. In the current version of the manuscript in lines 26-31, the results & conclusions were not discussed and presented in the abstract in detail. Especially the quantification of the results and numerical information of sensibility and specificity need to be incorporated.
  4. In the introduction, kindly incorporate more details and references to support your statements, especially the statements such as “pharmacogenetic evaluations for individualized antineoplastic drug response are not common in clinical practice” (in lines 60-61).
  5. In the introduction, very importantly please include the knowledge gaps existing in the current research work and prior studies performed in the field. Please specify the need for the current work presented in the manuscript.
  6. In the last paragraph of the introduction, kindly include the details of the broader impacts on the study made and the results achieved. It is very important to provide the future scope of the research performed to make a strong impact on the readers on the research performed.
  7. In the materials and methods (section-2), please include the figure with the schematic of the in-check lab on chips including the dimensions and the specifications of the chip design. Also please provide more details and specifications of the lab-on-chip design, in the explanation.
  8. In the Results section (Section-3), the plots and results in  figures-3 & figure-4 were listed and mentioned. Kindly incorporate the logical reasoning and scientific conclusions made using the plots. Also, please use the ongoing research results with appropriate references to support your arguments and statements.
  9. In the conclusions, please include the numerical details of the results and scientific conclusions driven from the results listed.
  10. Please revise the manuscript with English grammar. There are few places that the manuscript needs to be improved with respect to English writing.

Author Response

We have revised the manuscript as follow:

1-3) The abstract was revised following the kind suggestions of the Reviewer.

4-6) The introduction was revised following the suggestions of the Reviewer. In particular, the indicated statement was revised, the gap pf knowledge was discussed and new references included; the final purpose of the study was included.

7) A schematic description of the platform was included in the introduction with the new figure a showing the entire pipeline.

8) The results section was re-edited with new figure, considerations and references.

9) As suggested, the conclusions were revised.

10) As suggested, English grammar was revised.

I hope that we have adequately addressed all the concerns kindly raised by the Reviewers.

Round 2

Reviewer 2 Report

Dear authors,

Thank you for updating the manuscript with the recommended changes.